# Development of an Injection Molding Process for Long Glass Fiber-Reinforced Phenolic Resins

**DOI:** 10.3390/polym14142890

**Published:** 2022-07-16

**Authors:** Robert Maertens, Wilfried V. Liebig, Kay A. Weidenmann, Peter Elsner

**Affiliations:** 1Karlsruhe Institute of Technology (KIT), Institute for Applied Materials—Materials Science and Engineering (IAM-WK), Engelbert-Arnold-Str. 4, 73161 Karlsruhe, Germany; wilfried.liebig@kit.edu (W.V.L.); peter.elsner@ict.fraunhofer.de (P.E.); 2Fraunhofer Institute for Chemical Technology ICT, Joseph-von-Fraunhofer-Str. 7, 76327 Pfinztal, Germany; 3Institute for Materials Resource Management MRM, Hybrid Composite Materials, Augsburg University, Am Technologiezentrum 8, 86159 Augsburg, Germany; kay.weidenmann@mrm.uni-augsburg.de

**Keywords:** thermoset injection molding, phenolic molding compound, fiber length measurement, long fiber processing, plasticizing work, injection work, screw mixing element, process data acquisition

## Abstract

Glass fiber-reinforced phenolic resins are well suited to substitute aluminum die-cast materials. They meet the high thermomechanical and chemical demands that are typically found in combustion engine and electric drive train applications. An injection molding process development for further improving their mechanical properties by increasing the glass fiber length in the molded part was conducted. A novel screw mixing element was developed to improve the homogenization of the long fibers in the phenolic resin. The process operation with the mixing element is a balance between the desired mixing action, an undesired preliminary curing of the phenolic resin, and the reduction of the fiber length. The highest mixing energy input leads to a reduction of the initial fiber length *L*_0_ = 5000 μm to a weighted average fiber length of *L*_p_ = 571 μm in the molded part. This is an improvement over *L*_p_ = 285 μm for a short fiber-reinforced resin under comparable processing conditions. The mechanical characterization shows that for the long fiber-reinforced materials, the benefit of the increased homogeneity outweighs the disadvantages of the reduced fiber length. This is evident from the increase in tensile strength from *σ*_m_ = 21 MPa to *σ*_m_ = 57 MPa between the lowest and the highest mixing energy input parameter settings.

## 1. Introduction

### 1.1. Phenolic Molding Compounds and Their Applications

Phenolic resins are versatile polymers that are used in a variety of industrial and consumer applications. For example, they are used as binding systems for wood composites, paper, abrasives, and friction materials [1]. They also serve as matrix systems in fiber-reinforced composite materials, such as continuous fiber-reinforced laminates [2], long fiber-reinforced compression molding materials [3], and short fiber-reinforced injection molding compounds [4].

Typical applications for phenolic molding compounds that are established in the state of the art are oil pump housings, intake manifolds, valve blocks [4,5], and other small automotive parts, for example in the air condition systems [6] or turbochargers [7]. In these applications, the beneficial properties of the phenolic resin, such as high heat resistance, chemical resistance, and an overall excellent dimensional accuracy and stability come into play. Usually, the mechanical strength of the material plays a subordinate role for these parts. In the latest developments, phenolic molding compounds are used in electric motors [8,9,10], larger parts in combustion engines (such as camshaft modules) [11], and entire parts of crank cases [12]. In addition to the beneficial properties mentioned above, the mechanical requirements become more important due to the size and the structural nature of such large parts.

The most significant mechanical disadvantage of parts made from phenolic molding compounds are their low elongation at break and their high brittleness compared to thermoplastic polymers [4]. Long fiber reinforcement is especially beneficial for increasing the impact strength of a fiber-reinforced polymer material. This was proven by Gupta et al. [13], Thomason and Vlug [14], Rohde et al. [15], and Kim et al. [16] for glass fiber-reinforced polypropylene. Boroson et al. [17] conducted a study with glass fiber-reinforced phenolic resins with initial fiber length values in the molding compound ranging between L=3.5 mm and L=12.7 mm and found out that longer fibers significantly reduce the notch sensitivity during impact testing. However, they did not measure the residual fiber length in the molded part. Based on the literature data, it can be concluded that an attractive development aim is increasing the fiber length in parts manufactured from phenolic molding compounds to improve the impact toughness. 

Typically, there are two possibilities for increasing the fiber length in molded parts: First, using a semi-finished material, such as a long fiber granulate; or second, using a direct process in which longer fibers are incorporated into the final part. This decision-making process is a trade-off between the higher material costs of a semi-finished long fiber material and the more complex manufacturing processes of a direct process, most of which also involve a higher capital investment [18,19]. For fiber-reinforced thermoplastics, both process routes are well established and are available from multiple material and machinery equipment suppliers. However, in the case of phenolic molding compounds, neither long fiber-reinforced injection molding compounds nor established long fiber injection molding processes exist. Within this research paper, the development and the validation of a long fiber injection molding process for thermosetting phenolic resins is described.

### 1.2. Long Fiber Injection Molding Materials and Processes

Thermoplastic long fiber granulates are available in a variety of different lengths, typically ranging between s=6 mm and s=25 mm pellet length. Due to the pellet manufacturing process, the maximum initial fiber length, i.e., the fiber length before taking into account any process-induced fiber shortening, is limited to the size of the granulate [18]. The granulate size, in turn, is limited by the available dosing and feeding technology. For phenolic resin matrix systems, no long fiber granulates designed for injection molding are available on the market today. However, there are long fiber phenolic molding compounds for compression molding applications. They have a plate-like shape and are available in length classes of 5 mm, 12 mm, and 24 mm [20]. For compression molding applications, it is claimed that impact strength values that are 10 … 20 times superior to conventional short fiber phenolic molding [4], but these values were never reached in injection molding trials by Saalbach et al. [21] and Raschke [22].

Several process variants with direct fiber feeding were developed for thermoplastic materials [19,23,24,25,26,27,28,29,30,31,32,33]. In the injection molding compounder (IMC{ XE “IMC” \t “*injection molding compounder*”}), a co-rotating twin screw extruder is combined with an injection unit. The process was patented by Putsch [34] and first industrialized by KraussMaffei Technologies GmbH. Continuous roving strands are pulled into the extruder, which is coupled with the discontinuous injection process using a melt buffer. Besides the increase in fiber length, the main advantage of the inline compounding process is lowering the material costs. Due to the high capital investment costs and the responsibility for the material formulation, it is mostly only used for high-volume applications [26]. Another inline compounding process was developed by Composite Products, Inc. (CPI{ XE “CPI” \t “*Composite Products, Inc.*”}). It combines a continuous compounding process with a discontinuous injection process using a melt buffer [29]. The melting and compounding tasks are divided between two single-screw extruders. In contrast to the IMC process, the melt buffer and the injection unit are combined in one component. By using a check valve in the piston head, the compounding extruder can fill the backside of the piston head during the injection and holding phase. After the holding phase, the material can flow through the check valve to the other side of the piston head. 

By conducting the melting and compounding discontinuously, matching the injection molding cycle, no melt buffer is required. This reduces the capital investment costs and makes the direct compounding feasible for lower-volume applications. The direct compounding injection molding (DCIM{ XE “DCIM” \t “*direct compounding injection molding*”}) process, invented by Exipnos and KraussMaffei Technologies, couples a single screw compounding extruder with a traditional injection molding machine [30]. In the compounding extruder, the molder can tailor the material according to his needs by adding fibers, fillers, and other additives to the thermoplastic polymer. In the DIF{ XE “DIF” \t “*direct incorporation of continuous fibers*”} process (direct incorporation of continuous fibers), invented by Truckenmüller at the University of Stuttgart, continuous fibers are directly pulled into the screw of the injection molding machine [23,25]. Mixing elements on the injection molding screw are required for obtaining good fiber dispersion. The mechanical properties of the produced samples are comparable to conventional long fiber granulate. Another direct process for the injection molding of long fiber-reinforced thermoplastics was developed by Arburg GmbH & Co. KG in cooperation with SKZ Kunststofftechnik GmbH. In this process, called fiber direct compounding (FDC{ XE “FDC” \t “*fiber direct compounding*”}), the unreinforced thermoplastic granulate is passively pulled into the screw and melted, such as in a conventional injection molding machine [31,32,33]. In contrast to the DIF process, the continuous fibers are cut to a selectable length of L=2 mm … 100 mm using a fiber chopper and are fed to the injection molding machine via a twin screw sidefeed. Since the injection molding is a discontinuous process, the fiber feed is coordinated with the screw movement via the machine control system. At the position of the fiber feed, the screw core diameter is reduced to facilitate the incorporation of the fibers. 

### 1.3. Fiber Shortening and Fiber Length

Using the adhesion between fiber and matrix $\tau_{\mathrm{int}}$, the fiber diameter D, and the tensile strength of the fiber $\sigma_{F}$, the critical fiber length $L_{c}$ can be calculated according to Equation (1) [35].
(1)$$L_{c}=\frac{D\sigma_{F}}{2\;\tau_{\mathrm{int}}},$$
which is considered the minimum fiber length that is required for fully utilizing the reinforcement potential of the fibers. A fiber length ${L\;<\;L}_{c}$ still leads to a reinforcing effect, but does not fully utilize the available potential. Literature values for the critical fiber length $L_{c}$ in glass fiber-reinforced phenolic molding compounds vary between $L_{c}=2\;\mathrm{mm}$ [36] and $L_{c}=8\;\mathrm{mm}$ [37]. 

During the injection molding process, the fibers are subjected to high mechanical loads, causing fiber damage and breaking. Three distinct mechanisms for fiber shortening are identified [15,38,39,40]. First, fluid–fiber interactions are caused by viscous forces transferred from the polymer matrix into the fibers. For example, Gupta et al. [41] found in their study on the fiber length reduction of glass fiber-reinforced polypropylene, that a thin polymer film is initially formed on the surface of the screw and barrel wall when the matrix is melted. In this region, fibers that are anchored on one side in solid granulate are exposed to the shear flows of the molten polymer, which can lead to flexural failure of the fibers. According to their calculations, forces can occur that lead to fiber damage by buckling. Second, fiber–fiber interactions can be caused by fiber overlap. The amount of fiber–fiber interactions increases with increasing fiber content and increasing fiber length [42]. At the junction points of two overlapping fibers, the contact forces cause bending deformation of the fibers, which might lead to fiber breakage. Third, fiber–wall interactions happen at contact locations to machine parts. This is visible by the abrasive wear that can be found on the screw, the barrel, and other machine parts. 

Agglomerations and fiber bundles reduce the overall extent of the fiber shortening, resulting in a higher average fiber length compared to well-homogenized parts. Opening the fiber bundles works in the same way as breaking the fibers. Truckenmüller [43] investigated the opening of fiber bundles in the DIF process and concluded that fiber bundles can be treated as a single fiber with a larger fiber diameter and therefore a smaller L/D aspect ratio. This underlines the conclusion that fiber bundle opening is not possible without fiber shortening: Once the fiber bundle is opened, the aspect ratio of the individual fiber is significantly larger than the aspect ratio of the bundle from which the fiber originated. If the fluid forces are high enough for opening the fiber bundles, they likely will be high enough for shortening the individual fiber. An indicator for judging the existence of agglomerations and the degree of dispersion quality is the FLD ratio (fiber length distribution{ XE “FLD” \t “*fiber length distribution*”}) defined by Meyer et al. according to Equation (2) [44].
(2)$$\mathrm{FLD}=\frac{L_{p}}{L_{n}}$$

The average fiber length $L_{n}$ and the weighted average fiber length $L_{p}$ can be calculated according to Equations (3) and (4). $L_{i}$ is the length of the individual fiber i. The weighted average fiber length is the second moment of the fiber length distribution and is generally considered to be more descriptive because it has a higher emphasis on long fibers [45].
(3)$$L_{n}=\frac{\sum_{i=1}^{n}n_{i}L_{i}}{\sum_{i=1}^{n}n_{i}}$$
(4)$$L_{p}=\frac{\sum_{i=1}^{n}n_{i}L_{i}^{2}}{\sum_{i=1}^{n}n_{i}}$$

Meyer et al. calculated a theoretical value of FLD=1.44 for a fiber break in the middle due to viscous forces on the fibers. Once this value is reached, no further breakdown of the fibers due to fiber–fluid interactions shall occur.

### 1.4. Data Acquisition during the Injection Molding Process

With modern data acquisition technologies, the quantification of energy input into the polymer during plasticization and injection is possible. This is particularly important for reactive thermoset materials, such as phenolic resins. The screw torque during the plasticizing process $M_{\mathrm{Plast}}$ was monitored and analyzed by several authors. According to Rauwendaal [46], it is a good measure to quantify the mechanical power consumed by the extrusion process. For hydraulic injection molding machines, this plasticizing torque is typically calculated by measuring the pressure drop ${\Delta p}_{\mathrm{Hydr}}$ over the screw drive according to Equation (5).
(5)$$M_{\mathrm{Plast}}=\frac{\Delta p_{\mathrm{Hydr}}\eta_{\mathrm{Hydr}}V_{\mathrm{Drive}}}{20\pi },$$

Using the hydraulic efficiency $\eta_{\mathrm{Hydr}}$ and the hydraulic volume of the drive $V_{\mathrm{Drive}}$, Scheffler et al. identified an initial decrease in plasticizing torque with rising moisture content for phenolic molding compounds, followed by an increase towards very high moisture content values [47]. The fundamental softening effect of water in the polymer is the same for thermoplastics and thermosets, which explains the initial decrease in plasticizing torque. However, due to the lack of a non-return valve, further increasing the moisture content leads to a higher backflow during the injection phase for the thermoset molding compounds, and consequently a higher number of fully filled screw flights. In those fully filled screw flights, the molding compound is agitated and mixed during the screw rotation, leading to the rise in plasticizing torque [47]. In general, Scheffler [48] concludes that the plasticizing torque for thermosetting molding compounds is influenced by multiple factors, but has a strong correlation to the backflow during the injection phase. Several authors [49,50] used the injection work $W_{\mathrm{Plast}}$, which is the integral of the plasticizing power $P_{\mathrm{Plast}}$, as a measure for the total energy input into the polymer during the plasticization phase, see Equation (6).
(6)$$\begin{array}{c} W_{\mathrm{Plast}}=\int_{{\mathrm{Pl}}_{\mathrm{St}}}^{{\mathrm{Pl}}_{\mathrm{End}}}P_{\mathrm{Plast}}dt=\int_{{\mathrm{Pl}}_{\mathrm{St}}}^{{\mathrm{Pl}}_{\mathrm{End}}}\left(M_{\mathrm{Plast}}\times \omega \right)dt\\ =2\pi \int_{{\mathrm{Pl}}_{\mathrm{St}}}^{{\mathrm{Pl}}_{\mathrm{End}}}\left(M_{\mathrm{Plast}}\times n\right)dt \end{array}$$

In Equation (6), $M_{\mathrm{Plast}}$ is the plasticizing torque and n is the screw rotational speed. For a standard injection molding process using thermoplastic materials, Kruppa [49] observed an increase in the plasticizing work with increasing screw speed. This increased energy input leads to a stronger shortening of glass fibers, as described by Truckenmüller [43]. With increasing plasticizing work, fiber length asymptotically approaches a threshold value, which appears to be independent of initial fiber length and glass fiber content. A similar approach to quantify the energy input into the material during the injection phase of the process is the calculation of the injection work $W_{\mathrm{Inj}}$, which is the integral of the injection force $F_{\mathrm{Inj}}$ over the injection distance s according to Equation (7).
(7)$$W_{\mathrm{Inj}}=\int_{{\mathrm{Inj}}_{\mathrm{St}}}^{{\mathrm{Inj}}_{\mathrm{End}}}F_{\mathrm{Inj}}ds=A_{\mathrm{Piston}}\int_{{\mathrm{Inj}}_{\mathrm{St}}}^{{\mathrm{Inj}}_{\mathrm{End}}}p_{\mathrm{Hydr}.,\mathrm{Inj}}ds$$

Lucyshyn et al. [51], as well as Schiffers [52], use the injection work as a measure for viscosity changes of thermoplastic polymers during the process, e.g., due to a change in moisture content. A higher moisture content leads to a lower viscosity and consequently to a lower injection work. The injection work is also used as a control parameter for the injection process by several authors. Woebcken [53] described a method to compensate for changes in the material and/or the machine and mold setup by adjusting the screw movement during injection to reach a specific, previously defined injection work value. Cavic [54] used the injection work for judging the reproducibility of the injection molding process. All cited works deal with thermoplastic materials. The usage of the injection work to evaluate the curing state of the material in the thermoset injection molding processes is not yet reported.

As outlined above, no process for the direct feeding of long glass fibers into the injection molding process for thermoset resins in general and for phenolic resins in particular exists in the state of the art. In this paper, the development of such a process is described. An essential part of the process development and a significant addition to the state of the art is the application of a novel screw mixing element for the injection molding of fiber-reinforced phenolic resins. The machine process data are analyzed and used for the process development by calculating the injection and plasticizing work. By means of the structural and mechanical properties of the molded parts, the long fiber injection molding process is evaluated and compared to the state-of-the-art processing of short glass fiber-reinforced phenolic molding compounds.

The methods and the results that are presented within this paper were partially published in earlier publications by the authors. In publication [55], the method development for the fiber length measurement is described in detail. For the present paper, this measurement method is used to generate the fiber length distribution results. The results themselves are not published yet. The publication [56] describes the process development for the twin screw extruder compounding of the short glass fiber-reinforced phenolic molding compounds. The compounds that were manufactured according to this method constitute the basis for the new experimental investigations and process data analyses that are presented here. In small extracts, the results of the material characterization were presented at conferences [57,58]. This paper is a comprehensive presentation of both the process development and the material characterization results.

## 2. Materials and Methods

### 2.1. Materials

The phenolic molding compound used within this work is based on the Vyncolit^®^ X6952 short glass fiber-reinforced compound by Sumitomo Bakelite (Gent, Belgium). The material has a tailored composition of short glass fibers (SGF) and long glass fibers (LGF). The SGFs of the type DS5163-13P with a diameter of D=13 µm were sourced from 3B fibreglass (Hoeilaart, Belgium) and added to the molding compound in fractions between ϕ=0 wt.-% and ϕ=30 wt.-% by a twin screw extruder compounding on a lab scale extruder with a screw diameter of d=27 mm (Leistritz Extrusionstechnik GmbH, Nürnberg, Germany). The powdery resin components were melted in the first zones of the extruder by the barrel heating and by a screw kneading zone. Further downstream, the glass fibers were fed into the molten resin by using a sidefeed. A second kneading zone was used for opening the chopped fiber bundles. After leaving the extruder, the compound was cooled and granulated by using a cutting mill (Hosokawa Alpine, Augsburg, Germany). The method for controlling the energy input into the phenolic resin during the compounding, which was developed by the authors, is described in detail in the publication [56]. In publication [56], the detailed screw layout and the barrel temperature profile are presented. For the LGF, the Tt=2400 tex direct roving 111AX11 with a filament diameter of D=17 µm by 3B Fibreglass was used. The nomenclature of the material formulations follows the scheme PF-SGF*x*-LGF*x*. The variable *x* indicates the fiber weight content ϕ of the short or long glass fibers. Figure 1 gives an overview of the variations that were conducted. For selected material formulations, additional process and material variations were conducted. They are marked by the symbol and line styles in Figure 1. 

For this paper, the focus will be on material formulations with a total fiber content of $\phi_{\mathrm{tot}}\approx 30\;\mathrm{wt}.-\%$, which are positioned on the diagonal in the bottom left of Figure 1.

### 2.2. Long Fiber Thermoset Injection Molding Process

The long fiber thermoset injection molding process enables a flexible combination of SGF and LGF by separating the two mass flows (see Figure 2).

The SGF are gravimetrically fed as a part of the phenolic molding compound, whereas the LGF are chopped from the continuous rovings. Both mass flows are fed into the plasticizing unit with a twin screw sidefeed. The injection molding screw is a conveying screw with an interchangeable screw tip, which allows for the adaption of either a conventional conveying geometry or a newly designed, thermoset specific Maddock mixing element, which is shown in Figure 3a.

The mixing element is set apart from a conventional Maddock mixing element by three distinct geometrical features. First, the inlet channels have a gradual slope at their ends to facilitate the flow of material and to avoid material accumulations. Second, the thermoset Maddock mixing element has an edge fillet on the mixing flight, which results in additional elongational stresses on the material when passing through the shear gap. The third main feature is the reversed positioning of the mixing flight and wiping flight compared to the state of the art; traditionally, the pushing flank is also the wiping flank of the mixing element. Since thermoset molding compounds only start to melt in the foremost screw flights under the influence of screw flank pressure, the material close to the pushing screw flanks is molten, whereas the material distant from the screw flanks might still be granular [59]. If the molding compound entered a traditional mixing element in such a state, the granular fraction would be pushed through the shear gap, possibly blocking it. The thermoset-specific design ensures that only molten material enters the shear gap.

### 2.3. Injection Molding Parameters

During all injection molding trials with the long fiber process variants, rectangular plates with a size of 190 mm × 480 mm and a thickness of h=4 mm were molded. The plates were filled via a central sprue with a diameter of d=15 mm by using a KraussMaffei 550/2000 GX injection molding machine (KraussMaffei Technologies GmbH, Munich, Germany), which has a screw diameter of d=60 mm and a maximum clamping force of F=5500 kN. The basic specifications are given in Table 1. After molding, all plates were post-cured according to the temperature cycle in Figure 4.

### 2.4. Material Characterization

The test specimens were cut out of the molded and post-cured plates by waterjet cutting according to the cutting patterns shown in Figure 5a,b. A waterjet cutting machine, iCUT water SMART (imes-icore GmbH, Eiterfeld, Germany), was used.

The quasistatic mechanical testing was carried out according to the technical standards of DIN EN ISO 527-2 (tensile testing) [60] and DIN EN ISO 6603-2 [61] (instrumented puncture impact testing). After the mechanical testing, the fracture surfaces were analyzed by using a Zeiss Supra 55VP instrument. For the overview images on the left side of Figure 13, an acceleration voltage of U ≈ 10 kV and a working distance of WD ≈ 28 mm were used. The detail images on the left side of Figure 13 were obtained with U ≈ 3 kV … 5 kV and WD ≈ 7 mm.

The fiber length measurement procedure, as well as the validation investigations regarding fiber damaging and selectivity towards longer or shorter fibers, are described in detail by the authors in the publication [55]. In the first step of the measurement method, a circular sample with a diameter of d=25 mm was extracted from the molded plate using waterjet cutting. A typical weight for such a sample is approximately m=3 g. Subsequently, the phenolic matrix was removed by means of pyrolysis at T=650 °C for a duration of t=36 h under air atmosphere by using a LECO TGA 701 (St. Joseph, MI, USA). The ash residue, which solely consists of the dry glass fibers, was transferred into V=1.5 L distilled water, and a small amount of acetic acid was added to support the fiber dispersion. The suspension was subjected to t=2 min in an ultrasonic bath to open the fiber bundles. The fiber concentration in this suspension was too high for obtaining an analyzable image, which is why further dilution was necessary. By transferring the suspension into a dilution device for further down-sampling, this process can be conducted in a repeatable and controlled manner. The dilution device consists of a beaker glass with a capacity of V=4 L and an outlet tap with a diameter of d=10 mm attached to its side. A propeller stirrer keeps the fibers distributed homogeneously within the suspension. 

The dilution and sample taking process steps are accomplished by opening the outlet tap and refilling the beaker with distilled water. Once the desired degree of dilution was reached, measurement samples were taken through the outlet tap and transferred to a Petri dish, which was then analyzed using the FASEP { XE “FASEP” \t “*fiber length measurement system (no abbreviation)*”} system by IDM systems (Darmstadt, Germany). The cropping of the image and thresholding were done manually, but the fiber detection was done automatically using the algorithms provided by the FASEP system. Per Petri dish, approximately n=3000 (long fiber molding compound) to n=6000 (short fiber molding compound) fibers were measured. To reduce the influence of the variation in the sample taking, it was repeated at least four times per specimen. 

## 3. Results

### 3.1. Process Development

Figure 6 shows the injection pressure for a PF-SGF0-LGF30 formulation on the left side (a), and for a comparable short fiber formulation (PF-SGF28.5-LGF0) on the right side (b). Both materials were molded with both screw geometries. Using the conveying screw produces a pronounced pressure peak at the beginning of the injection stroke. The pressure requirement for the rest of the injection stroke is rather constant or slightly decreasing.

In contrast, the mixing element has a significantly lower initial pressure peak and a lower pressure requirement during the filling phase. Towards the end of the injection stroke, the pressure rises sharply until the switchover point to the holding pressure is reached. When comparing the two different plasticizing screw speeds n=40 1/min and n=70 1/min for the PF-SGF0-LGF30 material formulation and the mixing element (Figure 7a), the higher screw speed results in a tendentially lower initial pressure peak. During the remaining injection stroke, no clear distinction between the two screw speeds can be observed.

For the same material formulations as above, Figure 7 shows the screw position during injection and plasticizing. The total material throughput *Q*, which is adjusted by the peripheral devices (gravimetric loss-in-weight feeder and fiber chopper), approaches as close to the maximum possible feeding rate as possible, so that the plasticizing time is minimized.

For the long fiber formulation PF-SGF0-LGF30 in Figure 7a, the mixing element leads to a smoother screw movement with less scattering than the conveying geometry. To a less pronounced extent, this is also valid for the short fiber formulation PF-SGF28.5-LGF0. 

The characteristic values plasticizing work and injection work are used for the evaluation of the process stability and to determine the process limits of the long fiber direct injection molding process. In contrast to a conventional injection molding process, in which the screw is flood fed by pulling the granulate out of the material hopper, the long fiber direct injection molding process offers the possibility to starve feed the screw due to the adjustability of the material feeding rate. The material throughput, and therefore the plasticizing time, is defined by the mass flow provided by the gravimetric dosing of the granulate and the cutting speed of the fiber chopper. It is independent of the screw speed, which means that the screw speed can be used as a parameter for influencing the mixing quality and the energy input into the material. Figure 8 shows the plasticizing and injection work for a parameter study by using a PF-SGF0-LGF30 material formulation and the screw mixing element. Over the course of 13 injection molding cycles, the screw speed was increased from n=30 1/min to n=120 1/min. 

For each injection molding cycle, Figure 8 shows the plasticizing work and the corresponding injection work. For the first cycle 0, the material was plasticized in manual mode, which is why no plasticizing work was recorded by the machine. The increase in plasticizing work with increasing screw speed is clearly visible. Up to a screw speed of n=80 1/min, both work integrals remain stable at the respective screw speed increments. For the highest screw speed value *n=120 1/min*, the injection work rises despite a constant plasticizing work. The cycle 12 was the last moldable part of this parameter study. Despite reducing the screw speed to n=60 1/min after recognizing the instability of the process, the plasticizing work increased dramatically, and no injection was possible due to a curing of the material on the mixing element. 

Figure 9 shows the plasticizing work and the injection work of a process stability study using screw speeds of n=40 1/min and *n=70 1/min*. For both parameter combinations, respectively, 10 (trial number 1) and 9 (trial number 2) injection molding cycles were performed after a stable process was established.

For both screw speeds, the plasticizing work is stable. The effect of the increased screw speed on the plasticizing work and the injection work is in accordance with the values measured during the parameter study shown in Figure 8.

The plasticizing work was analyzed for both the mixing element and the conveying screw for several material formulations (see Figure 10). 

For the conveying screw geometry, a clear increase in plasticizing work with increasing fiber content is visible. This is valid for short glass fiber (SGF), long glass fiber (LGF) and combined (PF-SGFx-LGFx) material formulations. In contrast to the conveying screw geometry, no clear correlation between the fiber content and the plasticizing work can be drawn for the mixing element. The mixing element causes an overall significantly higher plasticizing work. Formulations containing long glass fibers require a significantly higher plasticizing work compared to formulations with only short glass fibers. 

As noted above, the allowable material throughput had to be adjusted based on the material formulation. Lowering the throughput was required for higher fiber contents and longer fiber lengths. With both factors, the apparent density of the granulate–fiber dry blend decreases and the shear energy input during plasticizing increases. This means that less material is pulled into the screw per screw rotation (apparent density), and at the same time, the screw rotational speed must be decreased (to keep the shear energy input of the mixing element in a controllable range and to avoid overheating). Both aspects result in an increase in plasticizing time. For long glass fiber materials with L=5 mm fiber length, up to ϕ=60 wt.-% is possible with the conveying element, whereas only ϕ=44.5 wt.-% can be molded with the mixing element before the plasticizing time exceeds the heating time. Plasticizing times that exceed the heating time are undesirable because of the long contact time of the machine nozzle to the hot mold.

### 3.2. Mechanical Properties

For evaluating the effect screw mixing element on the mechanical proper, the focus of this section will be on the formulations with a fiber content of ϕ=30 wt.-% and an initial long glass fiber length of L=5 mm. Figure 11 shows the tensile strength parallel (0°) and perpendicular (90°) to the flow of material.

Switching from the conveying screw geometry to the Maddock mixing element significantly increases the tensile strength for all formulations and for both specimen orientations. Increasing the plasticizing screw speed when using the mixing element leads to a further increase in tensile strength for the PF-SGF0-LGF30 formulation in 0° orientation. For the other formulations and orientations, the change in tensile strength with increasing screw speed is within the standard deviation of the measurement. For most material formulation and process parameter combinations, the scattering of the measurement results also increases when using the mixing element. While the positive effect of the mixing element on the tensile strength is clearly visible from the measurement results, it must be noted that the overall highest absolute strength value for the formulations with a fiber content of ϕ=30 wt.-% is still reached by the short fiber material PF-SGF28.5-LGF0.

The direct comparison of samples with a total fiber content of ϕ=30 wt.-% shows that the formulations containing LGF profit the most from using the mixing element (see Figure 12).

Both for the PF-SGF0-LGF30 and the PF-SGF16-LGF14 material, the puncture impact energy increases significantly when using the mixing element. An increase in screw speed with the mixing element has no significant effect; the average value of puncture impact energy decreases, but within the scattering of the measurement. Compared to the SGF material, both formulations that contain LGF have significantly higher puncture impact energy when using the mixing element. 

### 3.3. Scanning Electron Microscopy

Figure 13 shows a comparison between the PF-SGF28.5-LGF0 sample (conveying screw) and the PF-SGF0-LGF30 sample (with conveying screw and mixing element). The images show the fracture surface of unnotched Charpy impact test specimens with a specimen orientation parallel to the flow of material, e.g., the flow of material is oriented perpendicular to the plane of the image.

The SGF material in Figure 13a shows a skin and core layer structure with a predominant fiber orientation in the specimen direction (e.g., 0° to the flow) on the two surfaces of the specimen and a predominantly perpendicular fiber orientation (e.g., 90° to the flow of material) in the core layer. The fibers are pulled out of the fracture surface with some resin residues on the face sides of the fibers. The holes in the matrix created by pulling out the fibers are frayed and irregular. 

Comparing the LGF specimens manufactured with the conveying screw in Figure 13b to the specimens molded with mixing element in Figure 13c shows a strong reduction in the number of fiber bundles. The specimen manufactured with the conveying screw has a very inhomogeneous fracture surface with fiber-rich bundle regions and resin-rich regions where almost no fibers are present. By using the mixing element, the number of bundles is reduced significantly and a more homogeneous distribution of the fibers across the sample is achieved. Additionally, a skin and core layer structure becomes visible, which is not detectable with the conveying screw setup. The overall visual impression suggests that the fibers were shortened by using the mixing element.

Analyzing the detail SEM images on the right side of Figure 13b,c shows that the fiber surfaces are blank and smooth. Fibers that are pulled out of the fracture surface leave sharp and well-defined holes in the matrix. The mixing element improves the dispersion of the fibers and reduces the number of bundles, but no difference is observed regarding the resin residue on the fibers. It is remarkable that a high number of fibers have resin residues on their face sides; this indicates that the resin adhesion on the face sides is significantly stronger than on the fiber circumference. 

### 3.4. Fiber Length Measurement

The fiber length measurement results are shown in Figure 14 and Figure 15. For all measurements, the initial fiber length $L_{0}$, the weighted average fiber length in the part $L_{p}$, and the quotient ${\mathrm{FLD}=L}_{p}{/L}_{n}$ are given.

For PF-SGF0-LGF30, the weighted average fiber length in the molded part is reduced from $L_{p}=1103\;µm$ for the conveying screw tip to $L_{p,\;40\;1/\min}=809\;µm$ and to $L_{p,\;70\;1/\min}=571\;µm$ for the mixing element at n=40 1/min and n=70 1/min, respectively. The fiber length measurement results show that the frequency of fibers in the length classes from L=251 µm to L=750 µm is the highest for the mixing element with the high plasticizing screw speed. For all classes L > 751 µm, the conveying screw tip results in the highest fraction. The cumulative frequency curves confirm these results. The ratio ${\mathrm{FLD}=L}_{p}{/L}_{n}$ is also reduced when using the mixing element. 

When using a pure SGF compound, the fiber shortening is less pronounced. For both screw geometries, a fiber shortening of ${\Delta L}_{p}\;\approx \;80\;µm\ldots 100\;µm$ compared to the granulate takes place. No significant change in the ratio ${\mathrm{FLD}=L}_{p}{/L}_{n}$ occurs during the injection molding process.

## 4. Discussion

### 4.1. Process Development

It was possible to run a stable injection molding process with the developed thermoset-specific mixing element. In comparison to the conventional conveying screw geometry, the process value scatter, both during the injection and the plasticizing phase of the process, is reduced. The reduced process value scatter is especially visible for material formulations containing long fibers. It is assumed that the shear gaps of the mixing element reduce the backflow during the injection phase due to the flow resistance being higher, as compared to the standard conveying screw geometry. This assumption is in accordance with the findings of Kruppa et al. [62,63], who found that the small gaps of their mixing element act comparable or superior to a standard non-return valve. 

The increased plasticizing work input by the mixing element is visible from the reduced pressure peak at the beginning of the injection stroke, which is typically attributed to the cold plug in the nozzle. The lack of this pressure peak indicates that the material in front of the screw is hotter and consequently has a lower viscosity and a less pronounced cold plug. The increase in injection pressure requirement towards the end of the stroke can be attributed to the advanced chemical reaction progress, which in turn results in an increased viscosity. 

It is deducted that the effect of backflow prevention accomplished by the mixing element outweighs the backflow-increasing effect of the lower material viscosity. To investigate the lower process scattering with the mixing element during the plasticizing phase of the injection molding process, the back pressure and the screw position during plasticizing is analyzed (see Figure 16).

With the conveying screw geometry, the pressure that is required for melting and homogenizing the compound is applied by the injection molding machine’s hydraulic system. Especially for long fiber materials with a low apparent density, the injection molding machine’s hydraulic system had difficulties maintaining a constant back pressure. It is assumed that this scatter in back pressure resulted in the unsteady backward screw movement. When using the mixing element, the pressure for melting is applied geometrically by the reduction of the flow channel cross section in the shear gap of the element [64]. The back pressure of the injection molding machine is only applied to compact the already molten material in front of the screw. Consequently, the machine was able to maintain a much more stable back pressure level and a steadier backward screw movement.

Based on the parameter study findings, an upper plasticizing work limit of $W_{\mathrm{Plast}}\;\approx \;400\;\mathrm{kJ}$ was set. The available data does not allow for a sharp distinction between a stable and an unstable process. This plasticizing work limit corresponds to a screw speed limit of n ≈ 80 1/min … 90 1/min for a PF-SGF0-LGF30 formulation. 

From the plasticizing times that were recorded with the mixing element and the conveying screw, it is concluded that the short glass fiber formulations PF-SGFx-LGF0 can be molded up to a fiber content of ϕ=60 wt.-% with the mixing element. The underlying assumption for this conclusion is that a plasticizing time that exceeds the heating time is not acceptable. For long glass fiber materials with L=5 mm fiber length, up to ϕ=60 wt.-% is possible with the conveying element, whereas only ϕ=44.5 wt.-% can be molded with the mixing element. 

### 4.2. Structure and Properties

Switching from the conveying screw to the mixing element increases the plasticizing work input, which leads to a stronger shortening of the fibers. In this regard, no difference to thermoplastics was found. Most studies of thermoplastics state that the fiber length in the molded part decreases with increasing screw speed, which is represented by the increased plasticizing work. Moritzer and Bürenhaus [65] confirmed this for PP-GF, Lafranche et al. drew the same conclusions for PA66-GF [66,67] { XE “PA66” \t “*polyamide 6.6 with glass fibers*”}. As an exception to the general consensus, Rohde et al. [15] only found a slight, but not statistically significant shortening effect of the screw speed for PP-GF. While a stronger fiber shortening for higher plasticizing work values is observed for the long fiber formulations, this is not the case for the pure short fiber formulation PF-SGF28.5-LGF0. No significant additional shortening of the fibers compared to the conventional conveying screw is observed. Since the ratio FLD=Lp/Ln is also unaffected, it is concluded that the slight fiber shortening is caused by abrasive wear on the machine surfaces and not by breakage due to fluid forces. 

In the SEM fracture surface images, a reduction of fiber bundle size and count is observed when using the mixing element. The very small amount of resin residue on the fibers indicates a very weak fiber–matrix adhesion for the specimens manufactured in the long fiber direct process. For manufacturing those samples, the LGF of the type 111AX11 [68], with a 2400 tex roving size, were used. The fiber type was chosen based on recommendations by the fiber and the resin suppliers. In fact, the same fiber type 111AX in a 1200 tex size is used for manufacturing the commercially available long fiber granulate PF-SGF0-LGF55 [69]. For this reason, the weak fiber–matrix adhesion for all the long fiber specimens is surprising. The SEM investigations also show the typical three-layer setup of the parts, consisting of two skin layers with fibers oriented in the direction of the flow and a core layer with perpendicular fiber orientation [70]. This structure could be observed for the materials with a high homogeneity. If many bundles are present, no distinct skin and core structure is visible. Instead, fiber bundles with a predominantly perpendicular orientation characterize the structure.

It is remarkable that using the screw mixing element leads to a higher fraction of the skin layer, which consequently leads to a higher fraction of fibers oriented parallel to the direction of the material flow. The process development results show that the mixing element and the screw speed both increase the plasticizing work, i.e., the energy input into the resin during plasticization. This likely results in a further advance in the resin’s curing process. The formation of the skin layer happens by the incremental curing of the resin on the surface [71]. Englich [70] found out that higher mold temperatures and longer injection times result in a thicker skin layer, because the incremental curing on the mold surface is either quicker (higher mold temperature) or has more time (longer injection time). For this reason, the conclusion that a higher plasticizing work also causes a thicker skin layer is drawn, because the resin’s curing is progressed further, which consequently facilitates the incremental final curing on the hot mold surface. The skin layer fraction is also plausibly the reason why the mechanical properties of the SGF and LGF profit more in 0° orientation than in 90° orientation when using the mixing element.

The results of the puncture impact testing clearly show that the absorbed impact energy increases with increasing LGF content and is further improved by using the mixing element. The puncture impact specimens have a diameter of *d*=60 mm, which means that a large area is mechanically stressed during the testing. The LGF are likely better capable of distributing the load into this bigger area than the SGF. 

## 5. Conclusions

With the developed long fiber direct thermoset injection molding process, a significant increase in the fiber length in the molded parts was achieved. The average weighted fiber length was up to four times higher than the state-of-the-art short fiber-reinforced phenolic resins, resulting in values in the range of $L_{p}=500\;µm\;\ldots \;1100\;µm$. While this is a significant improvement, it remains significantly below the literature values for the critical fiber length of $L_{c}=2\;\mathrm{mm}\;\ldots \;8\;\mathrm{mm}$ for the combination of phenolic resin and glass fibers. A strong fiber shortening in the long fiber direct thermoset injection molding process from the initial fiber length $L_{0}=5000\;µm$ down to a weighted average fiber length in the range of $L_{p}=500\;µm\;\ldots \;1100\;µm$ was observed.

In terms of the specific processing challenges presented for the long fiber thermoset injection molding process, the plasticizing work input by the screw mixing element must be carefully controlled to avoid overheating and curing. The characterization results show that in the conflicting field between fiber dispersion and fiber length, the focus must be on the good fiber dispersion in the phenolic matrix. Every mechanical property characteristic that was investigated was increased by an improved material homogeneity, despite the accompanying fiber shortening. Besides the glass fiber length, other structural characteristics, such as the distinctiveness of the skin and core layer structure, are influenced by the mixing element. The additional mixing energy input leads to a clearer development of this structure, as well as to a tendentially higher skin layer fraction, which in turn leads to a higher fraction of fibers oriented parallel to the direction of the flow. 

The poor fiber–matrix adhesion reduces the expressiveness of some results. Despite those restrictions, it is concluded that the central key to good mechanical properties is a homogeneous distribution of the fibers in the phenolic resin matrix. This homogenization is more important than the fiber length. With the successful process development and the invention of the screw mixing element for thermoset molding compounds, future research studies will now focus on more detailed material and process parameter variations.

## Figures and Tables

**Figure 1 polymers-14-02890-f001:**
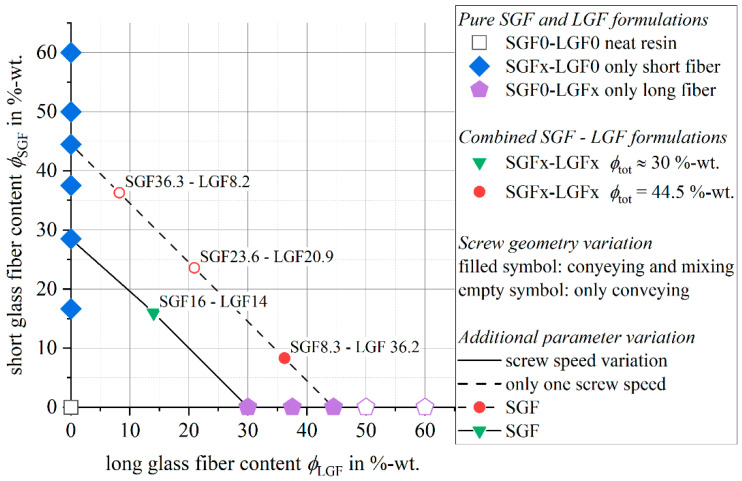
Material and process variations.

**Figure 2 polymers-14-02890-f002:**
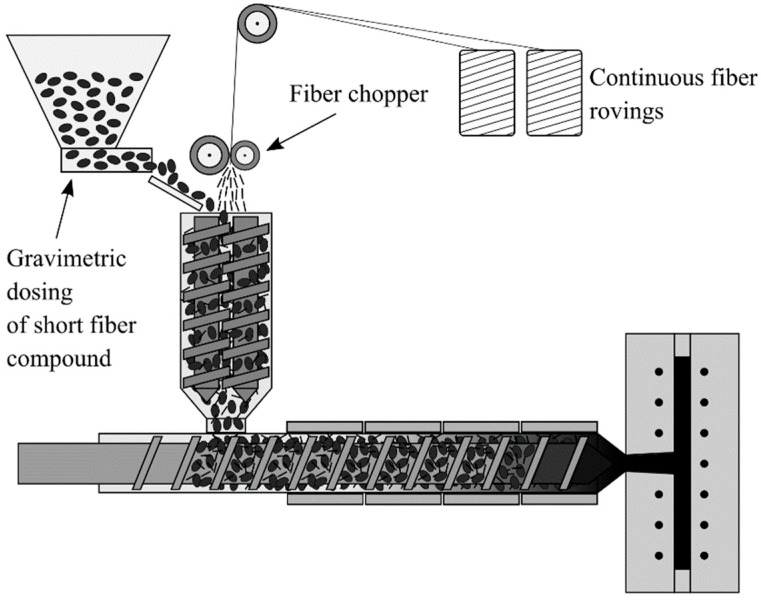
Process scheme of the long fiber thermoset injection molding process [57].

**Figure 3 polymers-14-02890-f003:**
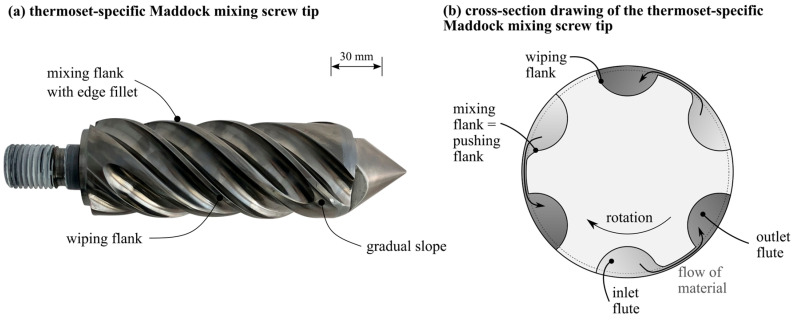
Thermoset-specific Maddock mixing element image (**a**) and cross section drawing (**b**).

**Figure 4 polymers-14-02890-f004:**
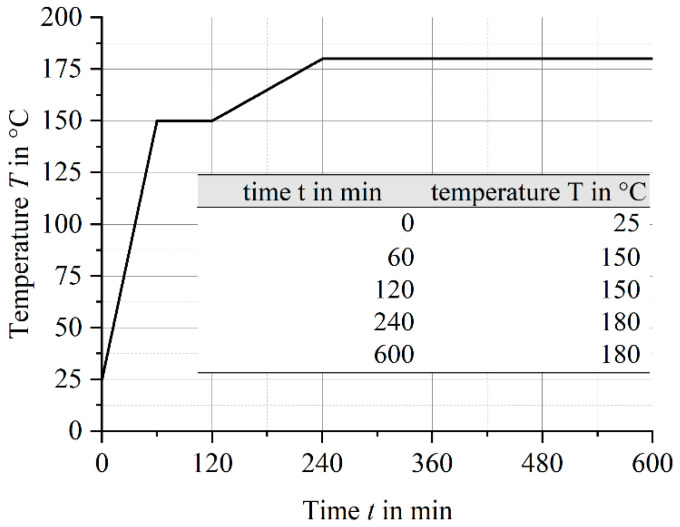
Post-cure cycle for molded plates.

**Figure 5 polymers-14-02890-f005:**
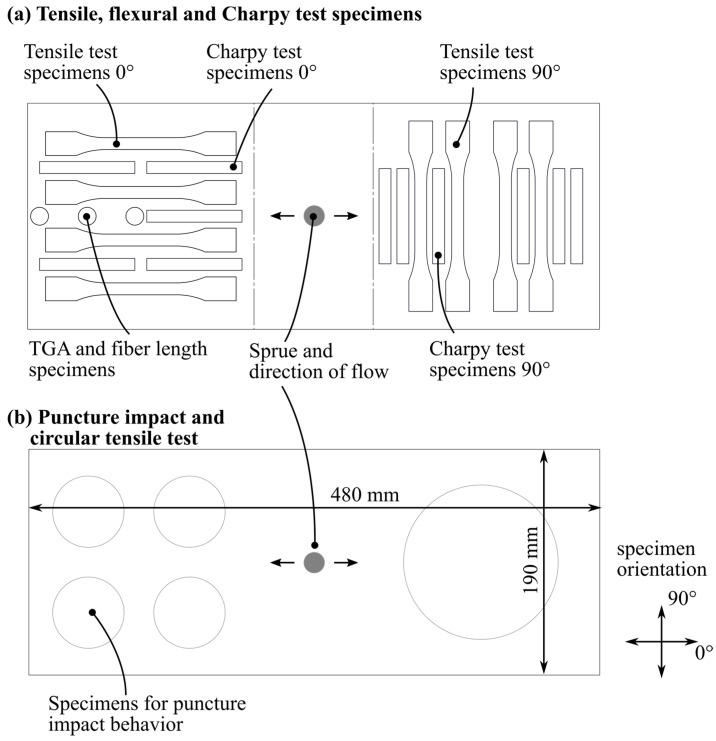
Cutting pattern for waterjet cutting of test specimens.

**Figure 6 polymers-14-02890-f006:**
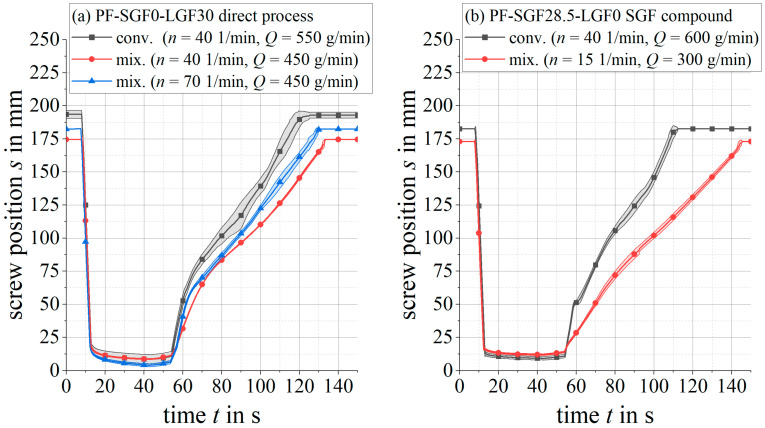
Injection pressure for LGF and SGF formulations with conveying and mixing screw.

**Figure 7 polymers-14-02890-f007:**
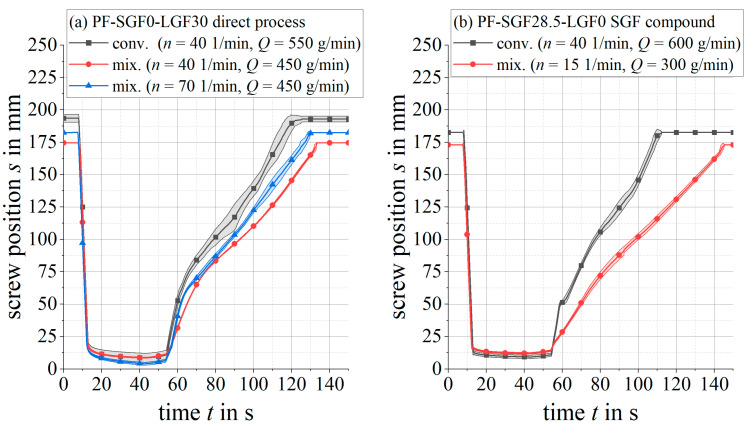
Screw position for LGF and SGF formulations with conveying and mixing screw.

**Figure 8 polymers-14-02890-f008:**
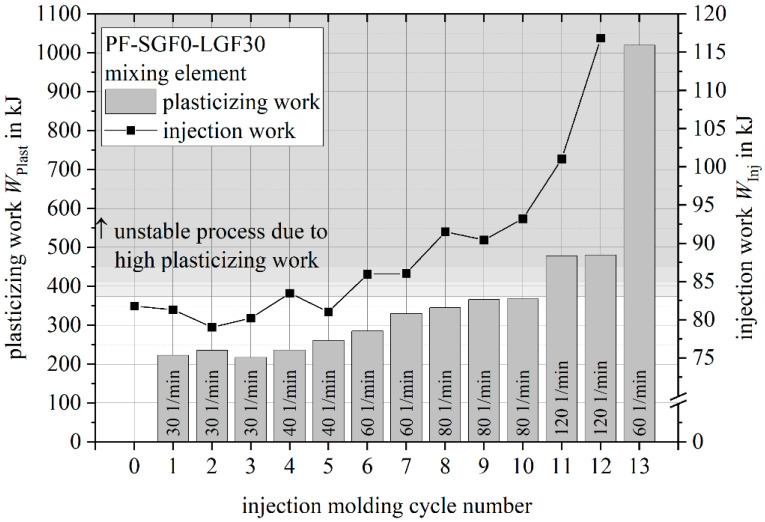
Plasticizing work and injection work for PF-SGF0-LGF30 screw speed study (usage and extension of the results presented in [57]).

**Figure 9 polymers-14-02890-f009:**
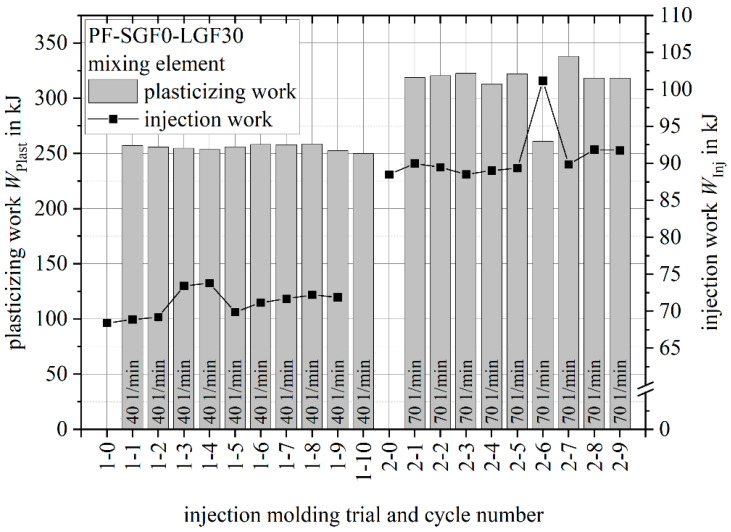
Plasticizing work and injection work for PF SGF0 LGF30 process stability study.

**Figure 10 polymers-14-02890-f010:**
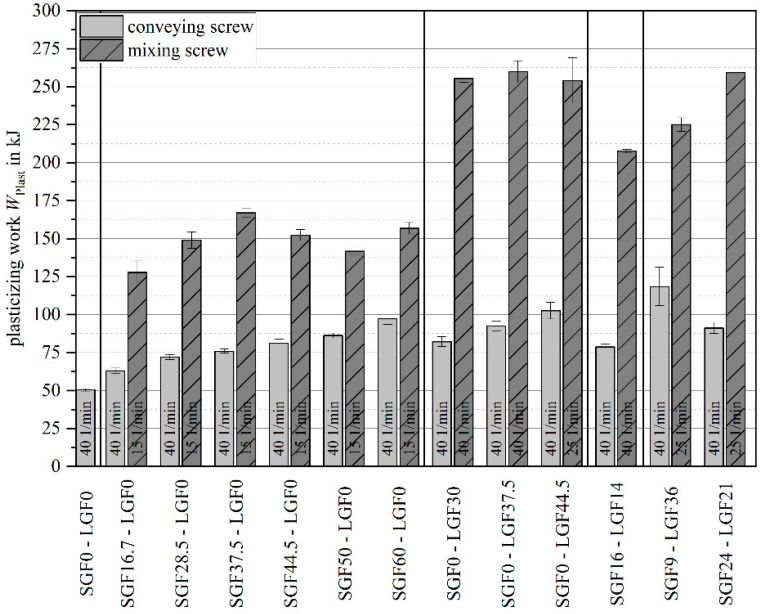
Plasticizing work for screw layout and material variations.

**Figure 11 polymers-14-02890-f011:**
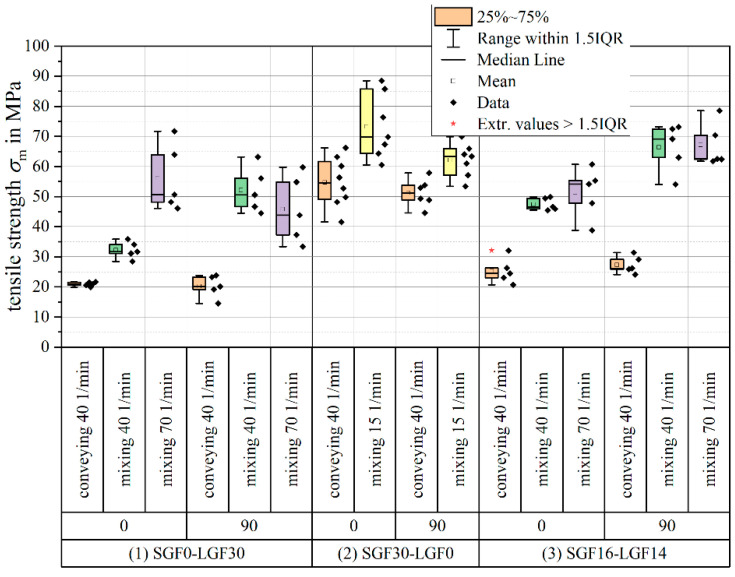
Tensile strength of 30 wt.-% specimens (usage and extension of the results presented in [57,58]).

**Figure 12 polymers-14-02890-f012:**
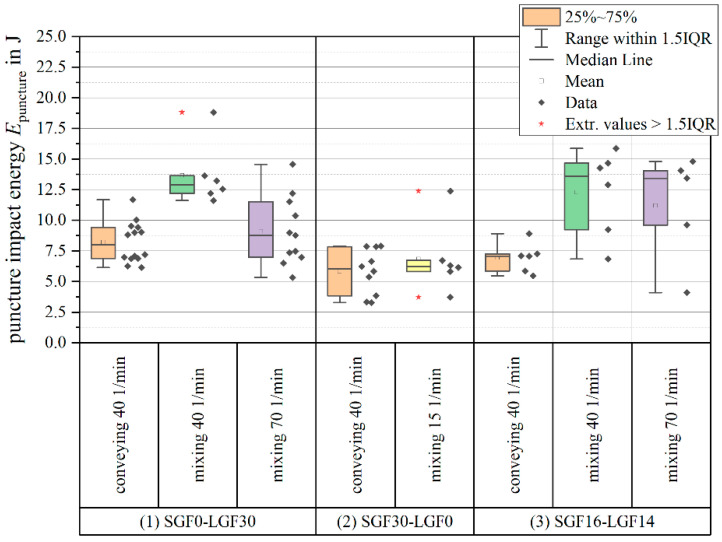
Puncture impact energy of 30 wt.-% specimens (usage and extension of the results presented in [57,58]).

**Figure 13 polymers-14-02890-f013:**
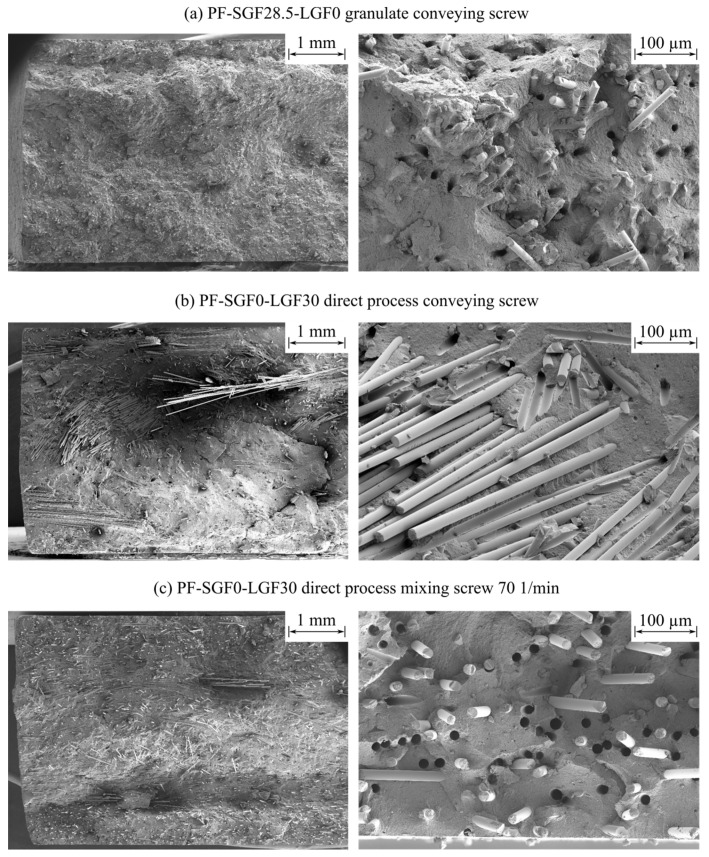
SEM for SGF granulate and LGF direct process mechanical testing specimens.

**Figure 14 polymers-14-02890-f014:**
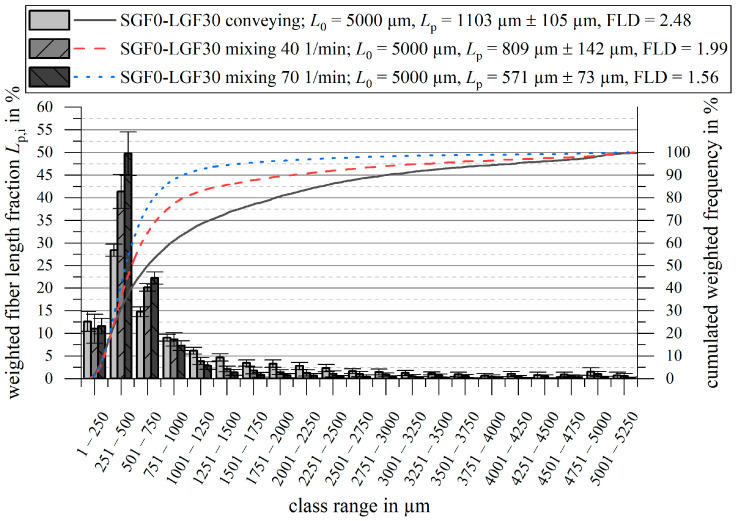
Fiber length measurement results for PF-SGF0-LGF30 (usage and extension of the results presented in [57,58]).

**Figure 15 polymers-14-02890-f015:**
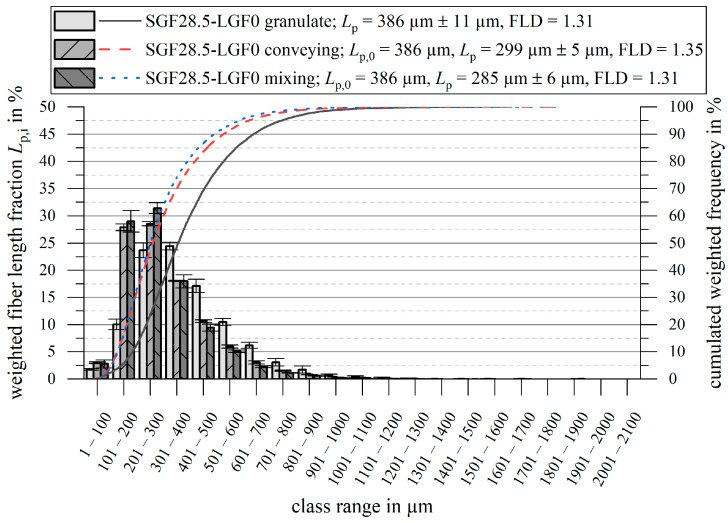
Fiber length measurement results for PF-SGF28.5-LGF0.

**Figure 16 polymers-14-02890-f016:**
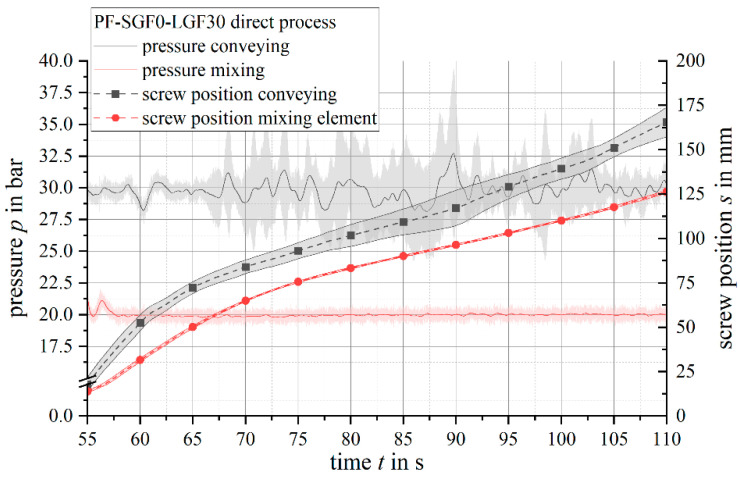
Pressure and screw position during plasticization of PF-SGF0-LGF30 material.

**Table 1 polymers-14-02890-t001:** Specifications of the KM 550/2000 GX injection molding machine.

Specification	Value	Unit
Screw diameter	60	mm
Max. plasticizing volume	792	cm^3^
Number of cylinder heating zones	4	-
Max. injection pressure	2420	bar
Max. injection speed	848	cm^3^/s
Clamping force	5500	kN

## Data Availability

The data presented in this study are available in the article.

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
