# Peer review of "Development of an Injection Molding Process for Long Glass Fiber-Reinforced Phenolic Resins"

_polymers, 2022, doi:10.3390/polym14142890_

Round 1
Reviewer 1 Report
The paper seeks to introduce an approach ‘’ Development of an Injection Molding Process for Long Glass Fiber-Reinforced Phenolic Resins”. However, the authors should consider improving upon the quality to further highlight and emphasize.
1. This is not a review work so consider adding some numbers from your results to the abstract.
2. In one or two lines, highlight the significance of the study.
3. The introduction needs to be improved by relating to the mechanics of the studied materials and their mechanical characteristics. The references to be included are: 10.1016/j.polymertesting.2017.09.009 and 10.3390/polym14132662.
4. The maximum number of words permissible in the keywords section is three. Consider reducing the words which exceed three to three words.
5. Consider removing “the” from “The suspension was the subjected to t = 2 min” in line 306. Or modify the sentence as “the suspension was then subjected to t = 2”.
6. Why was the ash residue in aqueous suspension used instead solutions like NaOH among others? Is there any other reason aside using it for the dilution?
7. What were the working range and the scale bar used in the SEM analyses?
8. Remove the word “there” in line 476.
Author Response
Reviewer 1 report and responses
Thank you very much for taking the time and for reviewing our manuscript so quickly. On behalf of all authors, I made changes to our manuscript and answered to your remarks below. The changes based on your feedback are marked in red font in the revised version of the manuscript. Changes made in response to reviewer 2 are marked in blue font.
Comments and Suggestions for Authors
The paper seeks to introduce an approach ‘’ Development of an Injection Molding Process for Long Glass Fiber-Reinforced Phenolic Resins”. However, the authors should consider improving upon the quality to further highlight and emphasize.
- This is not a review work so consider adding some numbers from your results to the abstract.
Based on your feedback, we added numbers for the results of the fiber length measurement and the tensile testing to the abstract. To stay within the word limit of around 200 words for the abstract, we carried out some changes in the first section of the abstract.
- In one or two lines, highlight the significance of the study.
In the penultimate paragraph of the introductory section, a sentence has been added to emphasize the importance of developing the new screw mixing element.
- The introduction needs to be improved by relating to the mechanics of the studied materials and their mechanical characteristics. The references to be included are: 10.1016/j.polymertesting.2017.09.009 and 10.3390/polym14132662
Thank you very much for the suggestion to improve the introduction section of the paper. We agree that highlighting the relation between the structure and the mechanical characteristics of the studied materials is important. To address your advice, we included two additional literature references and revised the third paragraph of Section 1.1. However, we decided against referring to the two articles you suggested and we would like to give a detailed explanation for this decision in the following parapgrahs.
In our manuscript, the quasi-static (tensile test) and dynamic (impact test) properties of short and long glass fiber-reinforced phenolic resins are studied. The suggested article “Influence of Stress Level and Fibre Volume Fraction on Fatigue Performance of Glass Fibre-Reinforced Polyester Composites” describes the fatigue behavior of a polyester resin reinforced with continuous glass fibers. This means that both the characterization method and the material system in this article are different from what we used for our manuscript. For this reason, we think that the suggested article is not relevant for our manuscript and we decided against referring to it.
The second suggested article “Experimental and modeling analysis of mechanical-electrical behaviors of polypropylene composites filled with graphite and MWCNT fillers” describes the addition of carbon nanotubes and synthetic graphene fillers to polypropylene. Our research work is centered around the addition of long and short glass fibers to a phenolic resin. This means that both the reinforcement / filler material and the matrix material are entirely different in our work compared to the suggested article. We are of the opinion that the suggested article is not relevant for our manuscript and we decided against referring to it.
Instead of the articles that you suggested, we added references to the following articles in the introduction section:
- Kim, Y.; Park, O.O. Effect of Fiber Length on Mechanical Properties of Injection Molded Long-Fiber-Reinforced Thermoplastics. Macromol. Res. 2020, 28, 433–440, doi:10.1007/s13233-020-8056-6.
- Gupta, V.B.; Mittal, R.K.; Sharma, P.K.; Mennig, G.; Wolters, J. Some Studies on Glass Fiber‐Reinforced Polypropylene.: Part II: Mechanical Properties and Their Dependence on Fiber Length, Interfacial Adhesion, and Fiber Dispersion. Polym. Compos. 1989, 10, 16–27, doi:10.1002/pc.750100104.
Both articles work with thermoplastics (polypropylene), but the manufacturing process (injection molding), the reinforcement material (long glass fibers) and the fundamental research question (influence of the fiber length on the mechanical properties of the molded parts) are relevant for our manuscript.
- The maximum number of words permissible in the keywords section is three. Consider reducing the words which exceed three to three words.
The “instructions for authors” website states the following requirements for the keywords in the front matter section: “Three to ten pertinent keywords need to be added after the abstract. We recommend that the keywords are specific to the article, yet reasonably common within the subject discipline.” Our understanding of your advice is to shorten the length of the individual keywords to three or less words, not to reduce the overall number of keywords. We carried out these changes in the revised version of the manuscript
- Consider removing “the” from “The suspension was the subjected to t = 2 min” in line 306. Or modify the sentence as “the suspension was then subjected to t = 2”.
Thank you for pointing out this grammatical error. As you suggested, we removed the word “the” from this sentence.
- Why was the ash residue in aqueous suspension used instead solutions like NaOH among others? Is there any other reason aside using it for the dilution?
Thank you for this question. The ash residue consists solely of the dry glass fibers. All the phenolic resin is removed during pyrolysis. This means that the only reasons for transferring the fibers into a suspension are dilution and dispersion, so that the optical fiber length measurement can be conducted. It is not necessary to dissolve any component. An aqueous suspension with a small amount of acetic acid serves the dilution and dispersion purposes, which is why other mediums such as NaOH were not considered.
Based on your feedback, we have made a small clarification in the relevant section of our manuscript.
- What were the working range and the scale bar used in the SEM analyses?
The scale bars are already present in each Subfigure of Figure 14. Based on your feedback, we added the image acquisition parameter working distance to Section “2.4 Material characterization” of the manuscript.
- Remove the word “there” in line 476.
We removed the word “there” and replaced it by “on the face sides”. The wording of the corrected sentence is now “It is remarkable that a high number of fibers has resin residues on their face sides, which indicates that the resin adhesion on the face sides is significantly stronger than on the fiber circumference.”
Reviewer 2 Report
The article "Development of an Injection Molding Process for Long Glass Fiber-Reinforced Phenolic Resins" is an engineering work describing a selected example of the influence of process parameters and the technology of adding filler on the process and the resulting structure and properties of composites. The work is extensive and complementary; despite a large amount of data, the authors clearly and logically presented them. The discussion, literature references, and criticism of the presented research results do not raise any major reservations. Despite the lack of a typical scientific character, the work may constitute an attractive source of knowledge for potential readers and supplement the knowledge available in the literature—the authors' advice is to complete the minor fixes listed below.
- Please correct the conclusion. In the first paragraph, the authors refer to the length of the fibers and write about significant improvements compared to literature data. Please conclude by how many more favorable effects are obtained even if estimated or percentage. You should also add the percentage decrease in length.
- Data from Table 1 should be presented in the text.
- Chart descriptions should be changed; please put units in appropriate brackets or decimals instead of describing with "in".
- The authors used a granulate containing short glass fibers; despite the reference to the previous work in which they described the shaping process, they should also include in this article brief information about the characteristics of the screws of the twin-screw extruder and the process parameters used to produce it.
Author Response
Reviewer 2 report and responses
Thank you very much for taking the time and for reviewing our manuscript so quickly. On behalf of all authors, I made changes to our manuscript and answered to your remarks below. The changes based on your feedback are marked in blue font in the revised version of the manuscript. Changes made in response to reviewer 1 are marked in red font.
Comments and Suggestions for Authors
The article "Development of an Injection Molding Process for Long Glass Fiber-Reinforced Phenolic Resins" is an engineering work describing a selected example of the influence of process parameters and the technology of adding filler on the process and the resulting structure and properties of composites. The work is extensive and complementary; despite a large amount of data, the authors clearly and logically presented them. The discussion, literature references, and criticism of the presented research results do not raise any major reservations. Despite the lack of a typical scientific character, the work may constitute an attractive source of knowledge for potential readers and supplement the knowledge available in the literature—the authors' advice is to complete the minor fixes listed below.
- Please correct the conclusion. In the first paragraph, the authors refer to the length of the fibers and write about significant improvements compared to literature data. Please conclude by how many more favorable effects are obtained even if estimated or percentage. You should also add the percentage decrease in length.
We reworked to first paragraph of the conclusion to address your feedback. This paragraph now contains number values for the initial fiber length and the fiber length in the molded parts manufactured in the long fiber direct thermoset injection molding process. Additionally, we added the previously cited literature values for the critical fiber length. We decided against expressing the fiber length decrease in percentage values. The number values are more informative and less prone to misunderstanding than a percentage value.
- Data from Table 1 should be presented in the text.
Thank you very much for your advice. We decided to provide the two main specifications of the injection molding machine, namely the screw diameter and the clamping force, directly in the text. The remaining specifications (plasticizing volume, number of cylinder heating zones, maximum injection pressure, maximum injection speed) are already listed in Table 1. To ensure the readability of this subsection of the manuscript and to avoid duplication between Table 1 and the text, we decided not to list them in the text.
- Chart descriptions should be changed; please put units in appropriate brackets or decimals instead of describing with "in".
Thank you very much for this advice. In our opinion, the most important aspect for a formality such as axis labeling is a consistent use of one labeling scheme throughout the entire publication. The axis labeling in our manuscript follows the scheme "name" "symbol" "in" "unit". It is in accordance with the German standard DIN 461:1973-03 “Graphical representation in systems of coordinates”. To the knowledge of the authors, no international standard for the axis labeling in charts exists. We used our chosen labeling scheme consistently for all Figures of the manuscript. The instructions for authors for the Polymers journal do not state a clear requirement for the format of the axis labeling, either.
Some recently published articles in Polymers use different ways for showing the unit in the axis labels of the charts:
- Zaghloul, M.Y.; Zaghloul, M.M.Y.; Zaghloul, M.M.Y. Influence of Stress Level and Fibre Volume Fraction on Fatigue Performance of Glass Fibre-Reinforced Polyester Composites. Polymers 2022, 14, 2662. https://doi.org/10.3390/polym14132662: Unit in round brackets
- Eckel, E.; Wiegel, K.; Schlink, A.; Ayeb, M.; Brabetz, L.; Hartung, M.; Heim, H.-P. Determination of Local Electrical Properties Using a Potential Field Measurement for Electrically Conductive Carbon Fiber Reinforced Plastics with Metal Contact Pins Joined via Injection Molding. Polymers 2022, 14, 2805. https://doi.org/10.3390/polym14142805: Unit described with “in”, exactly as we did in our manuscript
- Alrshoudi, F.; Abdus Samad, U.; Alothman, O.Y. Evaluation of the Effect of Recycled Polypropylene as Fine Aggregate Replacement on the Strength Performance and Chloride Penetration of Mortars. Polymers 2022, 14, 2806. https://doi.org/10.3390/polym14142806: Unit in round brackets
- Perera-Mercado, Y.; Hedayat, A.; Tunstall, L.; Clements, C.; Hylton, J.; Figueroa, L.; Zhang, N.; Bolaños Sosa, H.G.; Tupa, N.; Yanqui Morales, I.; Canahua Loza, R.S. Effect of the Class C Fly Ash on Low-Reactive Gold Mine Tailing Geopolymers. Polymers 2022, 14, 2809. https://doi.org/10.3390/polym14142809: Unit in square brackets
- Storck, J.L.; Ehrmann, G.; Güth, U.; Uthoff, J.; Homburg, S.V.; Blachowicz, T.; Ehrmann, A. Investigation of Low-Cost FDM-Printed Polymers for Elevated-Temperature Applications. Polymers 2022, 14, 2826. https://doi.org/10.3390/polym14142826: Unit in round brackets
While it seems that most publications use round brackets for stating the unit, there is no consistent style for the Polymers journal. In addition, there is no universal, internationally accepted standard. For these reasons, we decided to keep our labeling style according to the German standard DIN 461 and not to change the chart descriptions in the revised version of the manuscript. We kindly ask for your understanding for this decision.
- The authors used a granulate containing short glass fibers; despite the reference to the previous work in which they described the shaping process, they should also include in this article brief information about the characteristics of the screws of the twin-screw extruder and the process parameters used to produce it.
Based on your advice, we added a verbal description of the extruder screw layout and the post processing of the molding compound in Section “2.1 Materials” of the manuscript. With the additional information, the compounding process for obtaining the short fiber-reinforced granulate is now more comprehensible. We decided against the addition of another Figure (for example showing the screw layout and the temperature profile), because such a Figure is already included in our publication we are referring to.